# A New Approach for Determination of the Botanical Origin of Monofloral Bee Honey, Combining Mineral Content, Physicochemical Parameters, and Self-Organizing Maps

**DOI:** 10.3390/molecules26237219

**Published:** 2021-11-28

**Authors:** Tsvetomil Voyslavov, Elisaveta Mladenova, Ralitsa Balkanska

**Affiliations:** 1Department of Analytical Chemistry, Faculty of Chemistry and Pharmacy, Sofia University “St. Kliment Ohridski”, 1164 Sofia, Bulgaria; elimladenova@chem.uni-sofia.bg; 2Department of Special Branches, Institute of Animal Science, 2232 Kostinbrod, Bulgaria; r.balkanska@gmail.com

**Keywords:** self-organizing maps, botanical origin determination, physicochemical parameters, chemical elements, melissopalynological analysis

## Abstract

A new approach for the botanical origin determination of monofloral bee honey is developed. The methodology combines mineral content and physicochemical parameters determination with intelligent statistics such as self-organizing maps (SOMs). A total of 62 monofloral bee honey samples were analysed, including 31 linden, 14 rapeseed, 13 sunflower, and 4 acacia. All of them were harvested in 2018 and 2019 from trusted beekeepers, after confirming their botanical origin, using melissopalynological analysis. Nine physicochemical parameters were determined, including colour, water content, pH, electrical conductivity, hydroxymethylfurfural content, diastase activity, specific optical rotation, invertase activity, and proline. The content of thirty chemical elements (Ag, Al, As, B, Ba, Bi, Ca, Cd, Co, Cr, Cs, Cu, Fe, Ga, In, K, Li, Mg, Mn, Na, Ni, P, Pb, Rb, S, Se, Sr, Te, V, and Zn) was measured using ICP-OES, ICP-MS, and FAAS as instrumental techniques. The visualisation of the SOMs shows an excellent separation of honey samples in five well-defined clusters—linden, rapeseed, acacia, sunflower, and polyfloral honey—using the following set of 16 descriptors: diastase activity, hydroxymethylfurfural content, invertase activity, pH, specific optical rotation, water content, Al, B, Cr, Cs, K, Na, Ni, Rb, V, and Zn.

## 1. Introduction

Generally, honey is a multicomponent product, which contains, as its main substituents, carbohydrates (70–80%), water, small amounts of enzymes, proteins, organic acids, different chemical elements, vitamins, lipids, aromatic compounds, and biologically active substances. The mineral content of honey varies widely—from 0.02% in pale honey to 1.03% in dark honey [1]. The amount of all of the constituent components in bee honey depends on many factors, some of which are related to the type of bee that produced it (in Bulgaria, the bees *Apis mellifera* are breeding); others relate to the competence, techniques, and good practices of the beekeeper and honey producer. The type of plant from which the bees collect their nectar, and its ability to absorb minerals from the soil, and the climate and degree of the anthropogenic impact of the area, however, have the most noticeable impact [2]. The complexity of all of these factors makes choosing a methodical approach to identifying the different types of monofloral honeys by botanical origin and authenticity difficult [3,4].

Honey is a natural product that has been used as food and medicine since ancient times. It is produced by the processing of nectar by honeybees. Depending on its origin, honey is divided into monofloral (obtained predominantly from the nectar of one plant species) and polyfloral (obtained from different types of floral nectar). Different types of monofloral honey vary greatly in colour, aroma, and flavour. The physicochemical properties and honey composition of a certain kind of monofloral honey can also vary within itself, depending on seasonal weather conditions or geographical origin [5,6]. Monofloral honey, predominantly containing pollen from one plant species, is of higher quality, and is therefore more expensive than its polyfloral counterparts, which reinforces the need for an appropriate analysis technique, through which identifying the authenticity of monofloral honey would be made possible.

In order for a product to be labelled as honey, certain requirements, as described in Directive 2001/110/EC [7], such as the values of certain physicochemical parameters and/or pollen content of the honey, in order for it to be defined as monofloral [8,9], need to be met.

Systematic studies have been carried out on the ability to use a combination of different methods for proving the botanical origin of honey and potential fakes. The efforts are directed towards the selection of the appropriate descriptors, which allow for the preparation of a maximally authentic identification. It has been found that the determination of a single component on its own (chemical and physicochemical properties), such as a large number of chemical elements, hydroxymethylfurfural, water content, enzyme activity, mono- and disaccharides, nitrogen-containing compounds, or residues after antibiotic treatment of the bees, does not carry enough information on the geographical and botanical origin of honey [6]. On the contrary, determining the content of certain trace elements, and the ratios between the contents of certain elements, as well as the existence of a correlation between certain physicochemical parameters and the amounts of certain elements, are more reliable identifiers which could be used to classify different types of monofloral honey.

The application of these methods naturally requires the application of appropriate statistical analyses—cluster analysis [10,11,12,13,14,15,16,17,18,19,20,21,22,23,24], principal component analysis [11,12,13,14,16,17,21,22,23,25,26,27], etc. Combining the results of classical analytical methods for determining components in various types of monofloral honey with intelligent multivariate analyses, such as neural networks, is a new opportunity for the reliable identification of the botanical [17,20,22] and geographical [14,20,28] origin of monofloral honey. Kohonen self-organizing maps are one of these kinds of networks which are used for the botanical identification of different food products [29]; they could also be used for determining the botanical origin of different sorts of monofloral bee honey.

The bee honey which is produced in the territory of Bulgaria satisfies the needs of Bulgaria’s internal market. In addition, a big part of the production is exported to other countries in the European Union, as well as other countries outside of the European Union. This means that, regarding trade relations, honey production is an important part of the gross domestic product of Bulgaria. The bee honey produced in Bulgaria is mainly monofloral (e.g., from rapeseed, sunflower, thistle, linden, acacia, or coriander plants) [30]. The monofloral origin of the honey determines its higher price. Nowadays, consumers want to be sure about the definite origin of the products which they consume, due to which researchers are faced by a challenge—finding appropriate identifiers which allow for the reliable determination of fakes, false origin, and misleading statements about the authenticity of foods’ composition, which is an extremely difficult analytical task.

Traditionally, the botanical origin of monofloral honey has been ascertained by melissopalynological analysis—the pollen content is determined by microscopic examination, which is costly, time consuming, and requires personnel with high levels of expertise [8]. This in turn provokes the aim of this research—to develop a new methodological approach for the botanical origin determination of Bulgarian monofloral bee honey, which can be used as a suitable alternative to the traditional and expensive pollen analysis. This can be achieved by investigating the mineral content of traditional kinds of Bulgarian monofloral honey (e.g., linden, acacia, sunflower, coriander, lavender, thistle) and looking for a correlation or a common pattern between the mineral content, different physicochemical parameters, and the botanical origin of the honey.

## 2. Results

### 2.1. Samples

The bee honey samples used in the study were obtained from reliable producers, members of The Bulgarian Honey Producers Association, and were harvested in 2018 and 2019. Thirty-one linden honey samples were used—seventeen from 2018 and fourteen from 2019. Eight rapeseed samples from 2018 were used, and six from 2019. As a result of unfavourable weather conditions during the summer of 2018 in Bulgaria, a decline in sunflower honey harvesting was reported, which is why only thirteen such samples from 2019 were used. In addition, only four acacia samples were used—two from 2018 and two from 2019—as a result of the rainy spring seasons of the aforementioned years, adding up to a total number of sixty-two monofloral bee honey samples. The botanical origin of all of the collected samples was preliminarily stated by the beekeepers and confirmed (excluding two sunflower honey samples) through melissopalynological analysis. The honey samples were kept in dark, room temperature (20 to 25 °C) conditions, before conducting any analyses.

### 2.2. Basic Statistics

The data matrix, comprising all 39 variables (physicochemical parameters and minerals), and all honey samples, was constructed using the data presented in Appendix A. The raw data obtained from the four acacia honey samples are presented in Appendix A. The basic statistics (min, max, mean, and standard deviation values) of all of the monofloral bee honey samples (excluding acacia), separated by botanical origin (linden, sunflower, rapeseed), for the two monitored harvesting years (excluding sunflower), are presented in Appendix A (linden), Appendix A (sunflower), and Appendix A (rapeseed).

### 2.3. Set of Descriptors

Using the data for mineral content and physicochemical parameters of the analysed honey samples, and the technique of self-organizing maps, the optimal set of descriptors was constructed. The indicator used for deciding whether to include the variable (mineral or physicochemical parameter) in the optimal set of descriptors was the successful separation by botanical origin of the honey samples. The set of descriptors was made by using a step-by-step reduction approach of the variables, all the while following the successful separation rule. The HIT diagram, associated with the first SOM, including all variables, is presented in Figure 1. A very good separation of honey samples in four well-defined clusters is observed: linden, rapeseed, acacia, and sunflower.

The grouping of variable planes (Figure 2) shows five main clusters of correlated variables. Element B has a specific location and forms a cluster by itself. The elements Ca and Cs also form a cluster, while the other variables are grouped in three large clusters. This observation illustrates the similar distribution of the parameters within each group of the honey samples.

After the step-by-step reduction of the number of input parameters, while maintaining the good separation of the samples into four different clusters (Figure 3) by botanical origin, the set of descriptors was constructed. The set is formed of sixteen variables: Diast, HMF, Invert, pH, Rot, Water, Al, B, Cr, Cs, K, Na, Ni, Rb, V, and Zn.

### 2.4. Botanical Origin Classification

A new SOM was constructed (Figure 4) with an input layer of all honey samples—the “training” and “testing” sets and the set of sixteen descriptors.

The linden honey cluster is very well defined—the neurons of this cluster are populated with all linden honey samples from the training set and the six linden honey samples included into the testing group. The ten sunflower honey samples are also separated in another cluster, eight of which are from the training group, while the other two are from the testing group. The rapeseed honey samples form the third group of samples. This cluster is populated with eleven rapeseed honey samples from the training set, three rapeseed honey samples from the testing set, and two acacia honey samples from the training set. The last acacia honey sample from the training set, along with the acacia honey sample from the testing set, is in the neuron between the neurons of the rapeseed cluster and the sunflower cluster. The acacia honey samples from the training set are located in three different adjacent nodes. There is one more well-defined cluster, populated with two honey samples, described as polyfloral according to the pollen analysis.

## 3. Discussion

Shortly after collecting the honey samples, they were analysed using pollen analysis for the botanical origin confirmation and various methods and procedures were used for determining their physicochemical parameters and element content. One of the sunflower honey samples was dismissed, because the result for the diastase number was equal to seven, which is below the lower limit of eight [7]. According to the melissopalynological analysis, it was proved that two of the honey samples, claimed to be of sunflower origin by the honey producer, were not. The sunflower pollen content was established to be about 25% and 30% only, respectively, in the two samples, where according to legislation [9], the sunflower pollen must be equal to or greater than 40% in order for monofloral sunflower bee honey to be classified as such. In addition, about 30% of the pollen was established to be from the *Fabaceae* family. Based on these results, the two honey samples were redefined as polyfloral.

The ICP-MS results showed that the contents of Ag, As, Se, and Te were under the detection limit of the method (0.0001 µg/kg), which is why these elements were dismissed from the honey samples dataset.

As previously mentioned, there were only four acacia samples available for use—two from 2018 and two from 2019—which was not enough to perform deep chemometric analysis. However, three of the samples were included in the training set and one in the testing set of samples. The results cannot be accepted as reliable, but they could be a good starting point for future studies.

The procedure for constructing the set of descriptors started with training a SOM with an input layer of 47 confirmed monofloral bee honey samples and all 35 variables (9 physicochemical parameters and 26 elements). The HIT diagram (Figure 1) showed an excellent separation by botanical origin into four clusters. The elements B, Ca, and Cs were separated from all other variables into two different clusters. The element B was alone in its cluster and, due to this specific behaviour it was included in the final set of descriptors. The second cluster, populated with Ca and Cs, was also characterised by a specific location, which was a sign of a similar distribution of these parameters into the honey samples. The other clusters of variables were populated with many parameters showing a similar distribution.

The step-by-step reduction of variables started by dismissing a random parameter from the big clusters, e.g., Ca from a (Ca, Cs) cluster; Li from a (Al, Ba, Cd, Cu, Ga, K, Li, Sr, V, Invert, pH) cluster; Mn from the next one, and so on, until the last cluster. After every step of the reduction, the separation by the botanical origin of samples was checked. If the separation was unsuccessful, the reduction step was reversed, and a new variable was excluded from the input layer. This procedure was repeated tens of times until the formation of the final set of 16 descriptors.

The testing set of honey samples consisted of one acacia honey sample, six linden samples, two sunflower samples, three rapeseed samples, and two polyfloral honey samples, confirmed by the melissopalynological analysis. As a result, the linden, sunflower, and rapeseed honey samples from the testing set populated the linden, sunflower, and rapeseed clusters formed by the linden, sunflower, and rapeseed honey samples from the training set, respectively. Only the separation of the acacia honey samples was spoiled, which was attributed to the small number of acacia samples used in the study (four in total). One of the acacia training samples and the acacia sample from the testing set, ended up together in one neuron, while the other two acacia samples from the training set were included in two different neurons of the rapeseed cluster.

The polyfloral honey samples were separated in a different cluster unassociated with the other monofloral clusters.

## 4. Materials and Methods

### 4.1. Chemicals and Reagents

All reagents used were of analytical-reagent grade and all aqueous solutions were prepared with MilliQ water (MilliporeCorp., Milford, MA, USA). The stock standard solutions were CertiPUR^®^ ICP Multi-element Standard Solution IV (23 elements in diluted nitric acid), Merck Millipore, 1000 mg/L; and CertiPUR^®^ Reference material GF AAS Multi Element Standard Solution XVIII, Merck Millipore, 100 mg/L. The working standard solutions were prepared weekly through appropriate dilutions. Concentrated HNO_3_ (65%, Suprapur, Merck, Kenilworth, NJ, USA) was used for the wet digestion of honey.

### 4.2. Melissopalynological Analysis

The botanical origin of the honey samples was confirmed by melissopalynological analysis, also known as pollen analysis. The pollen analysis was carried out according to the procedures of national legislation [9,31].

### 4.3. Physicochemical Analysis

Nine physicochemical parameters including colour (Col), electrical conductivity (Cond), diastase activity (Diast), hydroxymethylfurfural (HMF), invertase activity (Invert), acidity (pH), proline (Prol), specific rotation (Rot), and water content (Water), were determined [29,32]. The colour of the honey was determined with a Lovibond Honey ColorPod, UK, and presented in mm Pfund. The parameters of water content, hydroxymethylfurfural content, electrical conductivity, specific optical rotation, invertase activity, and proline were determined according to the procedure in [32]. Diastase activity and pH were determined according to the methods described in the Bulgarian State Standard [31].

### 4.4. Analysis of Chemical Elements

#### 4.4.1. Sample Preparation Procedure

The preparation of the honey samples was performed according to the procedure reported by other authors [33]. Accurately weighed honey samples (1.0000 ± 0.0002 g) were placed in the PTFE vessels of a microwave digestion system (Ethos Easy, Milestone, Milan, Italy), and 9 mL of 65% HNO_3_ and 2 mL of 30% H_2_O_2_ were added. The vessels were then left for an hour to cool down, after which they were sealed and placed in the rotor of a microwave oven. The digestion was carried out following the program of 1200 W/20 min at 120 °C, 1200 W/15 min at 170 °C, and 20 min of ventilation. The vessels were cooled down to room temperature. Samples were quantitatively transferred into centrifuge tubes and filled up to 15.0000 ± 0.0002 g using an analytical balance with MilliQ water. Blank samples were passed through the whole procedure.

#### 4.4.2. Apparatus

The contents of thirty elements were measured using an inductively coupled plasma optical emission spectrometer ( Ultima 2, Jobin Yvon, Edison, NJ, USA) and an inductively coupled plasma mass spectrometer (ICP-MS 7900, Agilent, Santa Clara, CA, USA ): Ag, Al, As, B, Ba, Bi, Ca, Cd, Co, Cr, Cs, Cu, Fe, Ga, In, K, Li, Mg, Mn, Na, Ni, P, Pb, Rb, S, Se, Sr, Te, V, and Zn. Some of the elements (Al, Ba, Cr, Cu, and Fe) were analysed using both techniques, to ensure the validity of the results. Additionally, the results for Li and Rb were confirmed using flame atomic absorption spectrometry.

Inductively coupled plasma optical emission spectrometry (ICP-OES) was used for the determination of elements with a concentration in mg/kg. The optimized instrumental parameters are presented in Table 1 and Table 2.

External calibration by a multi-element standard solution was performed. The square of the correlation coefficients (*R^2^*) for all calibration curves was at least 0.998. Multi-element standard solution of Al, Ba, Ca, Cu, Fe, K, Mg, Mn, Na, S, Sr, and Zn (ICP Multi-element Standard Solution IV, CertiPUR, Supelco, Bellefonte, PA, USA) with an initial concentration of 1000 mg/L for each and single element standard solutions of P (*Trace*CERT, Supelco, Bellefonte, PA, USA) and S (*Trace*CERT, Supelco, Bellefonte, PA, USA) with initial concentrations of 1000 mg/L, were mixed and used for calibration after appropriate dilution to obtain the following concentrations: 1.0, 5.0, 10, 20, and 30 mg/L for P and S, and 0.2, 0.5, 1.0, 2.0, and 5.0 mg/L for the other elements. All solutions were prepared with MilliQ water (MilliporeCorp., Milford, MA, USA). For stabilization of the standard solutions, ultrapure nitric acid (≥69.0% HNO_3_, TraceSELECT, Honeywell Fluka, Charlotte, NC, USA) was used.

Inductively coupled plasma with mass spectrometric detection (ICP-MS) was used for determining chemical elements with a concentration in μg/kg. Table 3 presents the established instrumental parameters for reaching the maximum signal/noise ratio and meeting the requirements for reliable ICP-MS measurement. Table 4 presents the optimal conditions selected for ICP-MS measurements.

External calibration by a multi-element standard solution was performed. The square of the correlation coefficients (*R*^2^) for all calibration curves was at least 0.99. A multi-element standard solution of 30 elements (ICP Multi-element Standard Solution VI, CertiPUR, Supelco, Bellefonte, PA, USA) with an initial concentration of 10 mg/L for Ag, Ba, Cd, Co, Cr, Cu, Ga, Li, Ni, Pb, Rb, and Te, and 100 mg/L for As and Se, and single element standard solutions of Cs (*Trace*CERT, Supelco, Bellefonte, PA, USA) and In (*Trace*CERT, Supelco, Bellefonte, PA, USA) with initial concentrations of 1000 mg/L were mixed and used for calibration after appropriate dilution to obtain the following concentrations: 5.0, 10, 20, 50, and 100 µg/L for As and Se, and 0.5, 1.0, 2.0, 5.0, and 10 µg/L for the other elements. All solutions were prepared with MilliQ water (MilliporeCorp., Milford, MA, USA). For stabilization of the standard solutions, ultrapure nitric acid (≥69.0% HNO_3_, TraceSELECT, Honeywell Fluka, Charlotte, NC, USA) was used.

### 4.5. Chemometrics

The approach of self-organizing maps (SOMs) or Kohonen maps [34] is a member of the big family of neural networks. The big difference from other neural networks is that SOMs do not need a target output. The SOM consist of two layers: the input layer of neurons (nodes), formed from the real honey samples, presented as *k n*-dimensional vectors (where *k* is the number of honey samples and *n* is the number of input variables), and the output layer of neurons, arranged as a two-dimensional map. The output layer is the real SOM; every neuron is presented as an *n*-dimensional vector, where *n* is the same number as the number of input variables forming the input layer.

The algorithm is based on a “winner-takes-all” approach, where the “winner” is the neuron whose vector matches most closely to the input vector of a given sample. The winner node adjusts its vector weights to match the weights of the vector sample, while the neurons surrounding the winner are also modified to look more like the input vector. The occupation of the nodes with samples and the adjustment of the output vectors is repeated tens of thousands of times.

During the training process, after every iteration, the distances between the input and output data are calculated and compared to each other [34]. Finding the minimal distance marks the end of the training process. The trained map graphically presents the grouping of samples and the distribution of variables, the latter of which is visualised as easy to interpret 2D planes. Plane ordering is done after calculating the correlation between the input parameters. This ordering shows the relationship between the variables and could be used as a variable selection procedure when forming the optimal set of descriptors [35].

In this study, all calculations concerning SOMs were performed by the licensed computing platform MATLAB R2019a, with a free SOM Toolbox 2.0 [36].

Sixty-one honey samples were randomly separated into two sets. The first set of forty-seven samples (i.e., the training set) was used for the descriptors set selection, and the second set of fourteen samples (i.e., the testing set) was used for the testing of the classification process. The testing set consisted of one acacia honey sample, six linden honey samples, two sunflower honey samples, three rapeseed honey samples, and two polyfloral honey samples.

The set of descriptors was selected by reducing the number of input variables (minerals and physicochemical parameters) of the training samples, using the SOM approach, all the while maintaining the good separation by the botanical origin principle in the output layer. The reduction of the variables was performed one by one, tens of times, keeping the “golden rule” of a good separation by botanical origin. If a good separation was not achieved after excluding a given variable, the same parameter was included again, and another one was excluded instead. The reduction procedure was repeated until the final set of descriptors was formed, which gave the best separation by botanical origin and a minimum number of input variables.

Using the descriptors set and both sample sets (training and testing) as an input layer, a SOM was trained, where the samples of the training set were used to identify the nodes specific to a given botanical origin. The honey samples from the testing set, which fell into the node classified by botanical origin, were attached to the same botanical origin.

## 5. Conclusions

Four different kinds of monofloral bee honey (linden, rapeseed, sunflower, and acacia) from Bulgaria were successfully separated using a new approach. The presented method combines the determination of a set of sixteen descriptors—diastase activity, hydroxymethylfurfural content, invertase activity, pH, specific optical rotation, water content, Al, B, Cr, Cs, K, Na, Ni, Rb, V, and Zn—with the method of self-organizing maps. The physicochemical parameters included in the descriptors’ set are regularly determined, in order to ensure quality control assurance in honey as a food product. The determination of the selected chemical elements in the descriptors’ set is a routine practice, and requires common instrumental techniques, such as ICP-OES and ICP-MS. The new method performs faster and cheaper analyses than the well-known melissopalynological analysis.

## Figures and Tables

**Figure 1 molecules-26-07219-f001:**
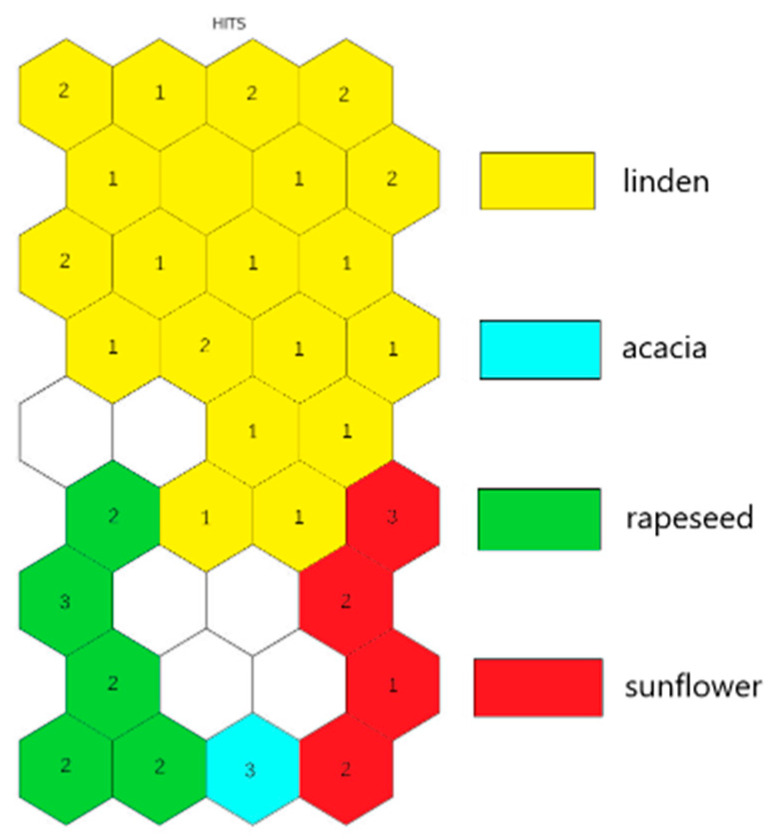
The HIT diagram of training set with all variables.

**Figure 2 molecules-26-07219-f002:**
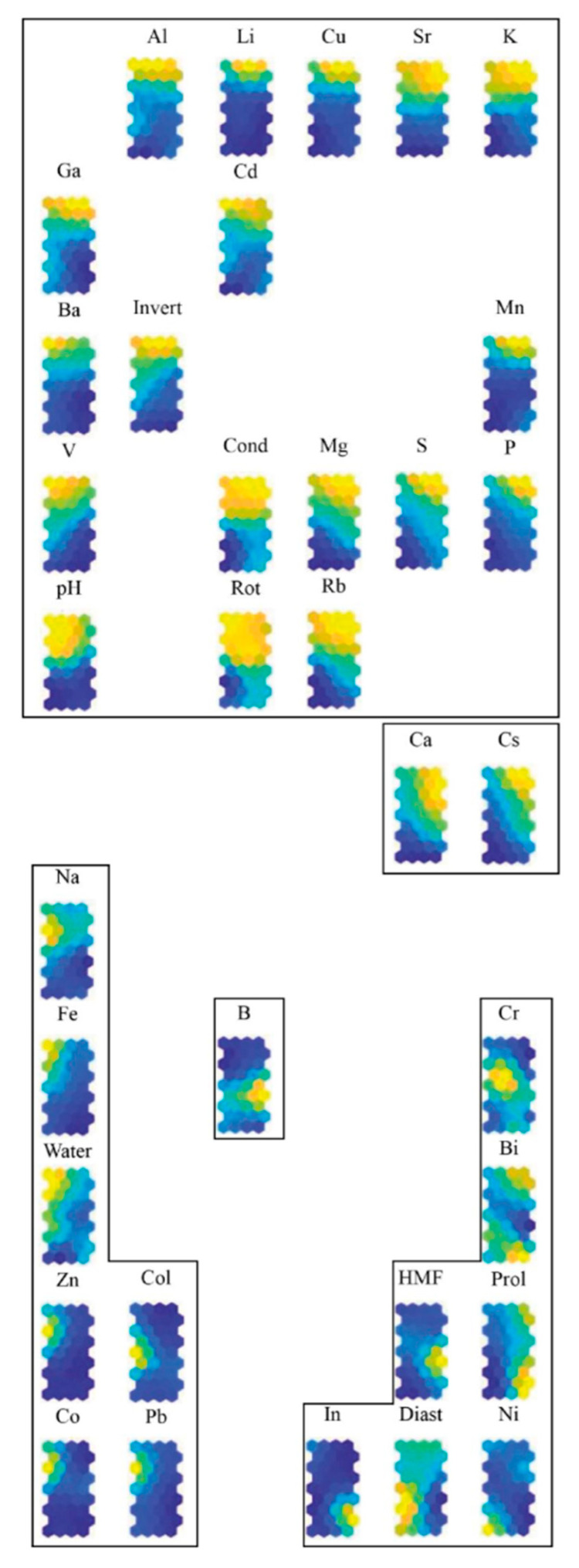
Ordering of variable planes in five main clusters. The abbreviations of the analytes are described in Section 4.3.

**Figure 3 molecules-26-07219-f003:**
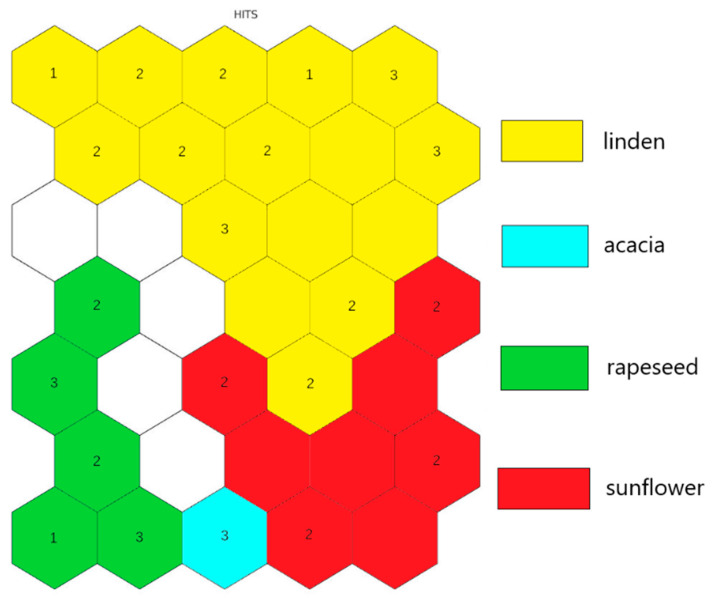
The HIT diagram of training set with the set of descriptors.

**Figure 4 molecules-26-07219-f004:**
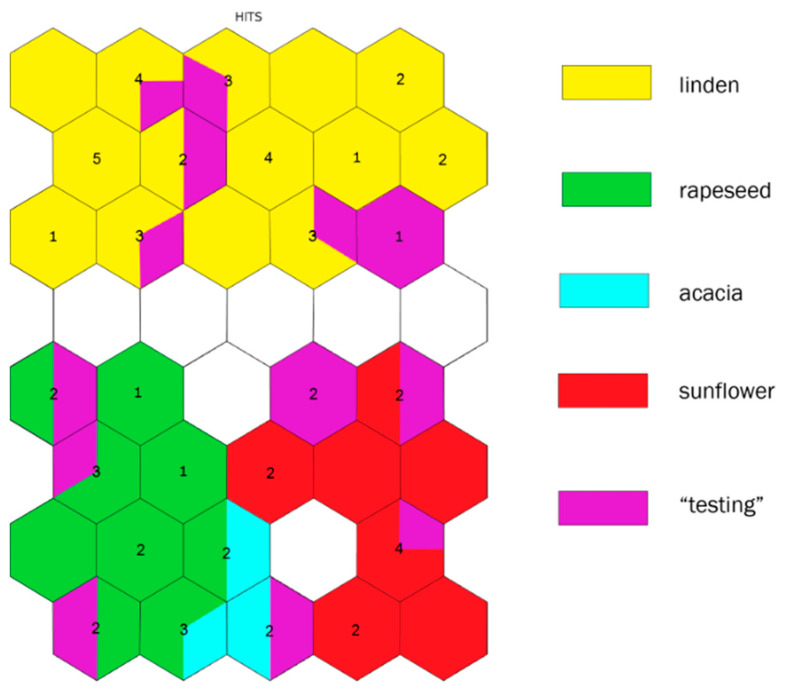
The HIT diagram of training and testing group of honey samples with the set of descriptors.

**Table 1 molecules-26-07219-t001:** Optimized instrumental parameters for ICP-OES measurements.

Generator power	1.00 kW
Type of nebulizer	V-groove
Plasma gas (argon)	15 L/min
Cooling gas	1.50 L/min
Gas of nebulizer	1.40 kPa
Observation time	5 s
Burner height	5 mm
Pump speed	30 rpm
Sample input time	12 s
Time for stabilization	15 s
Number of cues	5

**Table 2 molecules-26-07219-t002:** Spectral lines for measuring elements by ICP-OES.

Element	Spectral Line	Element	Spectral Line
Al	396.152 nm	Mn	257.610 nm
Ba	455.403 nm	Na	588.995 nm
Ca	370.602 nm	P	213.618 nm
Cu	324.754 nm	S	181.972 nm
Fe	238.204 nm	Sr	407.771 nm
K	766.491 nm	Zn	213.857 nm
Mg	279.553 nm		

**Table 3 molecules-26-07219-t003:** Instrumental parameters for ICP-MS measurements.

Generator power	1.20 kW
Type of nebulizer	Meinhardt (concentric)
Plasma gas (argon)	9 L/min
Additional gas	1.35 L/min
Nebulizer flow	1.1 L/min
Pump speed	20 rpm
Time for stabilization	5 s
Number of cues	5
Number of scans for cues	10

**Table 4 molecules-26-07219-t004:** Experimental conditions for ICP-MS measurement.

Element	*m*/*z*	Collision Cell Included	Isobaric Interference	Parallel Analysis
Li	7			The concentration is confirmed by FAAS
Al	27	Yes	^12^C^15^N^+^, ^13^C^14^N^+^, ^1^H^12^C^14^N^+^	The concentration is confirmed by ICP-OES
V	51	Yes	^34^S^16^O^1^H^+^, ^35^C^16^O^+^, ^38^Ar^13^C^+^, ^36^Ar^15^N^+^, ^36^Ar^14^N^1^H^+^, ^37^C^l1^4N^+^, ^36^S^15^N^+^, ^33^S^18^O^+^, ^34^S^17^O^+^	
Fe	57	Yes	^40^Ar^16^O^1^H^+^, ^40^Ca^16^O^1^H^+^, ^40^Ar^17^O^+^, ^38^Ar^18^O^1^H^+^, ^38^Ar^19^F^+^	The concentration is confirmed by ICP-OES
Co	59	Yes	^43^Ca^16^O^+^, ^42^Ca^16^O^1^H^+^, ^24^Mg^35^Cl^+^, ^36^Ar^23^Na^+^, ^40^Ar^18^O^1^H^+^	
Cr	52	Yes	^35^Cl^16^O^1^H+, ^40^Ar^12^C^+^, ^36^Ar^16^O^+^, ^37^Cl^15^N^+ 34^S^18^O^+^, ^36^S^16^O^+^, ^38^Ar^14^N^+^, ^36^Ar^15^N^1^H^+^, ^35^Cl^17^O^+^	The concentration is confirmed by ICP-OES
Ni	60	Yes	^44^Ca^16^O^+^, ^23^Na^37^Cl^+, 43^Ca^16^O^1^H^+^	
Cu	63	Yes	^31^P^16^O_2_^+^, ^40^Ar^23^Na^+^, ^47^Ti^16^O^+^, ^23^Na^40^Ca^+^, ^46^Ca^16^O^1^H^+^, ^36^Ar^12^C^14^N^1^H^+^, ^14^N^12^C^37^Cl^+^, ^16^O^12^C^35^Cl^+^	The concentration is confirmed by ICP-OES
Ga	71	Yes	^35^Cl^18^O2^+^, ^37^Cl^16^O^18^O^+^, ^37^Cl^17^O_2_^+^, ^36^Ar^35^Cl^+^, ^38^Ar^33^S^+^	
As	75	Yes	^40^Ar^35^Cl^+^, ^59^Co^16^O^+^, ^36^Ar3^8^Ar^1^H^+^, ^38^Ar^37^Cl^+^, ^36^Ar^39^K,	
Se	78	Yes	^40^Ar^38^Ar^+^, ^38^Ar^40^Ca^+^	
Rb	85			The concentration is confirmed by FAAS
Cd	111		^95^Mo^16^O^+^, ^94^Zr^16^O^1^H^+^, ^39^K_2_^16^O_2_^1^H^+^	
In	115			
Cs	133			
Ba	137			The concentration is confirmed by ICP-OES
Pb	208			
Bi	209			

## Data Availability

Not applicable.

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
