# Peer review of "A New Approach for Determination of the Botanical Origin of Monofloral Bee Honey, Combining Mineral Content, Physicochemical Parameters, and Self-Organizing Maps"

_molecules, 2021, doi:10.3390/molecules26237219_

Round 1

Reviewer 1 Report

There are some questions that need to be addressed:

Introduction.

  • The approach followed by the authors is not new. They must take into consideration the following works:  DOI: 10.1080/19440049.2017.1292054; DOI: 10.1080/19440049.2016.1239030).
  • The sentences reported in lines 91-111 and in lines 116-119 should be moved after line 40.
  • The sentences reported in lines 112-116 is a repetition of the concept reported in lines 28-32. Therefore, they should be deleted.
  • In my opinion, the Table 1 is not necessary.

Results

  • The caption of figure 2 is not exhaustive.

Discussion

  • There is the presence of some sentences of suggestions made to the authors (see lines 239-242 and 402-416). They should be deleted.
  • More sentences reported in discussion section overlapping with sentences reported in result section. A combined section results and discussion should be considered.

Author Response

Dear Reviewer,

We would like to thank you for your fast review of some important points, which we have used to improve the manuscript. Please find below the responses in detail to your comments.

We believe that we have addressed all of your concerns. To facilitate your review, we have copied all of the comments below, followed by our responses (in red).

We thank you for your assistance in this matter and we are looking forward to further contact.

Sincerely yours,

Tsvetomil Voyslavov

There are some questions that need to be addressed:

Introduction.

    •  

The approach followed by the authors is not new. They must take into consideration the following works: DOI: 10.1080/19440049.2017.1292054; DOI: 10.1080/19440049.2016.1239030).

According to our literature research, the approaches found and presented were a combination of the determination of different parameters (organic acids, volatile compounds, mineral content, antioxidants, etc.) and chemometric techniques as Principal Component Analyses, Cluster Analyses, Partial Least Squares Regression, but not Self-organizing maps. We agree to include the following work (DOI: 10.1080/19440049.2017.1292054), but the other one (DOI: 10.1080/19440049.2016.1239030) is not in the field of our research.

    •  

The sentences reported in lines 91-111 and in lines 116-119 should be moved after line 40.

Changed in the revised manuscript.

    •  

The sentences reported in lines 112-116 is a repetition of the concept reported in lines 28-32. Therefore, they should be deleted.

Changed in the revised manuscript.

    •  

In my opinion, the Table 1 is not necessary.

Deleted in the revised manuscript.

Results

    •  

The caption of figure 2 is not exhaustive.

Changed in the revised manuscript.

Discussion

    •  

There is the presence of some sentences of suggestions made to the authors (see lines 239-242 and 402-416). They should be deleted.

Deleted in the revised manuscript.

    •  

More sentences reported in discussion section overlapping with sentences reported in result section. A combined section results and discussion should be considered.

According to the Molecules Instructions for Authors, the “Discussion” section and the “Results” section must be separated, and the “Results” section must be before the “Discussion” section. We have prepared the manuscript according to the Instructions.

Reviewer 2 Report

The manuscript deals with a proposal for a new way to identify the plant species visited by bees from the pollen found in honey.
In fact, from my own experience, this stage of the study (palynological) is very expensive. 

The text needs to be more detailed, especially in the results and discussion (which need a lot of evolution).
Please note that ICP is an expensive equipment, available in few laboratories, so the advantages of its use should be better explained.

The differences in soil, water regime, nutrients, among others, can directly alter the content of these minerals and end up leading to a wrong result?

Some other observations are in PDF.

Author Response

Dear Reviewer,

We would like to thank you for your fast review of some important points, which we have used to improve the manuscript. Please find below the responses in detail to your comments.

We believe that we have addressed all of your concerns. To facilitate your review, we have copied all of the comments below, followed by our responses (in red).

We thank you for your assistance in this matter and we are looking forward to further contact.

Sincerely yours,

Tsvetomil Voyslavov

The manuscript deals with a proposal for a new way to identify the plant species visited by bees from the pollen found in honey.
In fact, from my own experience, this stage of the study (palynological) is very expensive.

One of the main reasons, for the very high price, is the need of high qualified specialists.

The text needs to be more detailed, especially in the results and discussion (which need a lot of evolution).

Changed in the revised manuscript.
Please note that ICP is an expensive equipment, available in few laboratories, so the advantages of its use should be better explained.

The ICP-OES and ICP-MS spectrometers are not so rare in the laboratories anymore. The determination of a set of few elements could be executed by a lot of members of the lab staff, in contrast to the melissopalynological analysis, where highly qualified specialists are needed.

The differences in soil, water regime, nutrients, among others, can directly alter the content of these minerals and end up leading to a wrong result?

This is true, and that is the reason to collect samples of one botanical origin but with different geographical origin (different regions of Bulgaria) from different years.

Some other observations are in PDF.

13 lines and only 2 citations? Strange...

Changed in the revised manuscript.

fakes???? it´s correct?

Yes, it is correct! The “fake” is a synonym of “counterfeit”.

This table isn't on journal standart.

Deleted in the revised manuscript.

where's reference???

Added in the revised manuscript.

reference?????

Added in the revised manuscript.

please, improve this Figure.

Changed in the revised manuscript.

Reviewer 3 Report

Review on manuscript no 1464510

A new approach for determination of botanical origin of monofloral bee honey, combining mineral content, physicochemical parameters and self-organizing maps

by Tsvetomil Voyslavov, Elisaveta Mladenova and Ralitsa Balkanska

submitted to Molecules

In the manuscript submitted for comments the Authors proposed new method for determination botanical origin of monofloral bee honey, base on minerals content, physicochemical parameters and self-organizing maps.

The topic raised by the authors is still valid and important, so such research results should be published, but the manuscript is inaccurate and in its current form I cannot recommend it for publication in the Molecules journal.

Detailed recommendations:

Table 1 – does not add anything interesting to the publication, so it should be removed,

line 105 – should be: research,

line 106 – why only for Bulgarian honeys?

lines 112-119 – the introduction should end with a clearly defined research goal resulting from the literature review, this general information about honey should be found in advance,

lines 122-125 and 130-134 – unnecessary repetition from the material section,

lines 136-144 – unnecessary repetition from the methods section,

lines 146-151 – the description of the obtained results collected in tables S1-S4 is missing,

line 183 – the abbreviations used in the figure should be explained in the caption,

line 206 – incorrect font,

lines 239-241 – these are editorial guidelines and should not be included in the manuscript,

lines 309-315 – descriptions of the methods used should be precise in order to be recreated, or should be accompanied by references to relevant literature or methodological standards,

line 322 – model, producer and origin country should be added,

line 338 – origin country should be added,

line 339 – more information about calibration curves should be given, e.g. origin of standards, number of measuring points, concentration ranges, values of the coefficient of determination for the curves,

Tables 2 and 4 – does not add important interesting to the publication, so it should be removed, some of this information can be provided in the text,

line 348 – origin country should be added,

lines 402-415 – these are editorial guidelines and should not be included in the manuscript,

Conclusion – should be extended,

line 501 – the title should be added, like in position 30.

Author Response

Dear Reviewer,

We would like to thank you for your fast review of some important points, which we have used to improve the manuscript. Please find below the responses in detail to your comments.

We believe that we have addressed all of your concerns. To facilitate your review, we have copied all of the comments below, followed by our responses (in red).

We thank you for your assistance in this matter and we are looking forward to further contact.

Sincerely yours,

Tsvetomil Voyslavov

Review on manuscript no 1464510

A new approach for determination of botanical origin of monofloral bee honey, combining mineral content, physicochemical parameters and self-organizing maps

by Tsvetomil Voyslavov, Elisaveta Mladenova and Ralitsa Balkanska

submitted to Molecules

In the manuscript submitted for comments the Authors proposed new method for determination botanical origin of monofloral bee honey, base on minerals content, physicochemical parameters and self-organizing maps.

The topic raised by the authors is still valid and important, so such research results should be published, but the manuscript is inaccurate and in its current form I cannot recommend it for publication in the Molecules journal.

Detailed recommendations:

Table 1 – does not add anything interesting to the publication, so it should be removed,

Deleted in the revised manuscript.

line 105 – should be: research,

Changed in the revised manuscript.

line 106 – why only for Bulgarian honeys?

The differences in soil, water regime, nutrients, among others, can directly alter the content of the chemical elements and end up leading to wrong results. That is the reason to collect samples only from Bulgaria – a country with a relatively small area. Our new approach could be applied for honeys with different geographical origin (countries).

lines 112-119 – the introduction should end with a clearly defined research goal resulting from the literature review, this general information about honey should be found in advance,

Changed in the revised manuscript.

lines 122-125 and 130-134 – unnecessary repetition from the material section,

The description of honey samples is done only in this section, on lines 122-125 and 130-134. In the “Material and Methods” section, we do not describe the sample collection, the origin of samples, or their number.

lines 136-144 – unnecessary repetition from the methods section,

Changed in the revised manuscript.

lines 146-151 – the description of the obtained results collected in tables S1-S4 is missing,

Edited in the revised manuscript.

line 183 – the abbreviations used in the figure should be explained in the caption,

Changed in the revised manuscript.

line 206 – incorrect font,

Changed in the revised manuscript, according to the Molecules template.

lines 239-241 – these are editorial guidelines and should not be included in the manuscript,

Deleted in the revised manuscript.

lines 309-315 – descriptions of the methods used should be precise in order to be recreated, or should be accompanied by references to relevant literature or methodological standards,

The physicochemical parameters determination was provided according to “Harmonized methods of the European Honey commission” [new reference number 32] and “Bulgarian State Standard – BDS 3050” [new reference number 31]. These two references were mentioned in the original manuscript.

line 322 – model, producer and origin country should be added,

Changed in the revised manuscript.

line 338 – origin country should be added,

Added in the revised manuscript.

line 339 – more information about calibration curves should be given, e.g. origin of standards, number of measuring points, concentration ranges, values of the coefficient of determination for the curves,

Added in the revised manuscript.

Tables 2 and 4 – does not add important interesting to the publication, so it should be removed, some of this information can be provided in the text,

The selected optimal instrumental parameters, which are presented in Tables 1 and 3 (according to the new table numbering) are very important for analyses recreation.

line 348 – origin country should be added,

Added in the revised manuscript.

lines 402-415 – these are editorial guidelines and should not be included in the manuscript,

Deleted in the revised manuscript.

Conclusion – should be extended,

Extended in the revised manuscript.

line 501 – the title should be added, like in position 30.

The title of this State standard is only “Bee honey”, nothing more. https://bds-bg.org/en/project/show/bds:proj:22569

Round 2

Reviewer 2 Report

The manuscript has been improved sufficiently.
I have no further considerations. It can be accepted for publication.

Reviewer 3 Report

After reassessing the manuscript: molecules-1464510-peer-review-v2, entitled "A new approach for determination of botanical origin of monofloral bee honey, combining mineral content, physicochemical parameters and self organizing

maps", by Tsvetomil Voyslavov, Elisaveta Mladenova  and Ralitsa Balkanska, I can state that authors made necessary corrections taking under consideration my recommendations. In my opinion, the quality of the manuscript has improved significantly, therefore it can be accepted for publication in the Molecules journal. Before publication, the article still requires technical correction regarding the correct layout of the text, figures and tables. The manuscript should also be proofread by a native speaker.